# Endodontic Rotary Files, What Should an Endodontist Know?

**DOI:** 10.3390/medicina58060719

**Published:** 2022-05-27

**Authors:** Ana-Belén Dablanca-Blanco, Pablo Castelo-Baz, Ramón Miguéns-Vila, Pablo Álvarez-Novoa, Benjamín Martín-Biedma

**Affiliations:** Unit of Dental Pathology and Therapeutics II, School of Medicine and Dentistry, University of Santiago de Compostela, 15705 Santiago de Compostela, Spain; pablocastelobaz@hotmail.com (P.C.-B.); ramon.miguens.vila@gmail.com (R.M.-V.); pablo.alvarez.novoa@hotmail.com (P.Á.-N.); b.martinbiedma@gmail.com (B.M.-B.)

**Keywords:** design, endodontics, NiTi, rotary files, rotation

## Abstract

Clinicians should be aware of all the characteristics and capacities of the instruments that are possible to use when conducting a root canal treatment. The wide variety of nickel–titanium (Ni-Ti) rotary systems on the market and the lack of standardisation of this type of instrument makes the choice in each specific case difficult. Therefore, this review is intended to summarize the characteristics that should be taken into account when choosing one instrument over another. It will be essential to know characteristics, of alloy from which the instrument is made. Moreover, the geometry of the instrument will determine its behaviour, being the mass, the one that marks its resistance to a greater extent. The movement performed by the file is another of the fundamental keys to understand rotary instruments. In conclusion, when performing root canal treatment, the characteristics of the instrument and the tooth must be taken into account, and the operator’s own limitations should be known. This paper provides the key points to keep in mind when making this type of treatment.

## 1. Introduction

The manufacture of manual endodontic instruments in stainless-steel is standardised under ISO regulations [1]; however, Ni-Ti rotary instruments do not have a single standard to comply with, so manufacturers design the geometry of the active part of the instrument in a non-standardised way. These designs seek to provide advantages in the instrumentation of root canals; however, the wide variety of systems on the market could be considered a disadvantage. The main problem in deciding which file to use in each case is often not knowing what to look for and why. Despite all the information that exists in the literature, it is difficult to find a review in which all the main information on rotary instruments is grouped. Therefore, this review aims to be a quick guide that summarizes what we must know and take into account to choose the best endodontic instrument in each case.

Despite the advantages of rotary NiTi systems, there is a consensus among researchers stating that the main disadvantage of rotary NiTi systems is the unexpected fracture of the instruments [2], directly influencing the prognosis of root canal treatment. However, some of these accidents could be avoided.

For all these reasons, this study aims to review the concepts and knowledge that all professionals preparing to perform endodontic treatment should bear in mind: the characteristics of rotary instruments, nickel (Ni)—titanium (Ti) alloy, advances in NiTi alloys, instrument design, continuous and reciprocate rotation, and instrument fracture.

## 2. Material and Methods

This narrative review was performed according to PRISMA GUIDE [3]. A literature survey was conducted to identify the main articles that sought to explain the main characteristics of the rotation instruments and their relationship with his clinical use. The research was carried out in Pub-Med using the keywords: rotation endodontic instruments, nickel–titanium alloy endodontic, and fracture endodontic instruments. The eligibility criteria were: complete articles published in the English language, full text available, experimental studies, clinical studies, and reviews. Exclusion criteria were given outside of English and unpublished, conference articles, and letters to the editor (Figure 1).

Two reviewers analysed all titles and abstracts of the articles found, independently and in duplicate. Articles that did not meet the inclusion criteria were excluded. In case of disagreement between reviewers, it was resolved through debate, aiming to seek the best correlations between the characteristics of a rotation endodontic instrument and its clinical use. We did not intend the present study to be a systematic review, but a narrative review, synthesizing the field of endodontics, providing an updated and relevant overview on the subject.

## 3. Review

### 3.1. Nickel–Titanium Rotary Instrument Characteristics

TAPER: The amount the file diameter increases for each millimetre along its working surface from the tip toward the file handle [4]. For example, a 25-gauge file with a taper of 0.2, would have a diameter of 0.27 to 1 mm from the tip and 0.29 to 2 mm from the tip.DIAMETER: A straight line that joins two points of a circle (in this case the cross-section of the file), passing through its centre. This may vary along the length of the entire file due to taper. Knowing the diameter of the file allows the professional to know the size of the file at the point of curvature, and with it the relative stress that is exerted on the file [4].CROSS-SECTION: The geometric shape that the file presents when it is sectioned perpendicular to the longitudinal axis, determines the amount of mass of the file.EDGE (Figure 2): The part of the file in contact with the dentin, which is the junction between two grooves.RADIAL LAND (Figure 2): If instead of a cutting edge, a plane is formed, the result is a radial land. Radial lands were designed to reduce the tendency of the file to screw into the canal, reduce transportation of the canal and limit the depth of cut. This design causes the file to abrade rather than cut, requiring more torque and more time to be effective when working.GROOVES OR FLUTES (Figure 2): The part of the file used to collect soft tissue and dentin chips removed from the wall of the canal. Its role is important in the cutting efficiency of the file; large grooves will allow more cutting than small grooves because they will take longer to saturate.HELICAL ANGLE (Figure 2): The angle formed by the cutting edge with the longitudinal axis of the file. It is responsible for the file cutting by rotation or traction and for evacuating the debris lodged in the grooves. It can be constant or variable, influencing the file’s degree of screw-in, therefore, it is decisive with respect to the torsion of the instrument.RAKE ANGLE: In the cross-section, this is the angle formed by the leading edge and the radius of the file. If the angle formed by the leading edge and the surface to be cut is obtuse, the rake angle is said to be positive or cutting. If the angle formed by the leading edge and the surface to be cut is acute, the rake angle is said to be negative or scraping.PITCH: Distance between corresponding points within which the pattern is not repeated. The smaller the pitch, the more spirals the file has and the greater the helix angle.

### 3.2. Nickel–Titanium Alloy: Key Characteristics

The chemical composition of most nickel–titanium alloys used in root canal treatment corresponds to 55-Nitinol and contains approximately 56% (wt) nickel (Ni) and 44% (wt) titanium (Ti) [5]. Ni-Ti has an inherent ability to alter its type of atomic bonding which causes unique and significant changes in the mechanical properties and crystallographic arrangement of the alloy. NiTi alloys contain three microstructural phases [5,6,7]:(a)Austenitic phase: Also called the high-temperature phase or parent phase. The NiTi alloy is in this phase at room temperature [6]. The austenitic phase is characterised by having an elastic behaviour or the ability of the material to recover its initial condition after yielding the force that causes deformation.(b)Martensitic phase: Also called low-temperature phase because the NiTi alloy is in this phase when the temperature is low [6]. The martensitic phase is characterised by having a plastic behaviour, that is, after the cessation of the force that causes the deformation, the alloy maintains said deformation [6].(c)R phase or pre-martensitic phase: The arrangement of the atoms in this phase is rhomboidal [6,8,9].

It is important to know the different microstructural phases since the NiTi alloy will have different mechanical properties depending on the phase in which it is found [10]. This ability to alternate between phases is what gives the alloy its two characteristic properties: superelasticity and shape memory [5].

#### 3.2.1. Shape Memory: Phase Change by Temperature

At a relatively high temperature (100 °C) the NiTi alloy is in the austenitic phase. If the alloy is cooled, from a certain temperature the crystals of the alloy change to the martensitic phase up to a certain temperature at which all the crystals of the alloy have become martensitic. The temperature at which this phenomenon begins is called the martensite transformation start temperature (Ms). The temperature at which martensite is completely transformed is called the martensite transformation finish temperature (Mf) [5,6].

The same thing happens in the reverse direction, such that the temperature at which this phenomenon begins is called the austenite transformation start temperature (As) and the temperature at which this phenomenon is complete is called the austenite finish temperature (Af) [5,6,11] (Figure 3). That is, the alloy does not change in the microstructural phase block, but there is a range of temperatures at which the alloy is transforming, and it is not completely in the martensitic phase or in the austenitic phase, but in a combination of both. This range of temperatures is called the Transition Temperature Range (TTR) or the Reverse Transition Temperature Range (RTTR), depending on the direction of the transition. The R-phase is located in a very narrow band in this range of transition temperatures.

This microstructural change produced in the alloy due to the change in temperature occurs by a process of sliding atoms. No macroscopic changes are observed during the phase transformation, but if the alloy is in the martensitic phase and we apply an external force, it can be easily deformed. The deformation can be reversed by heating the alloy above the TTR (or RTTR) resulting in the properties of the alloy returning to the previous properties. Thus, the martensitic transformation due to temperature change gives the alloy its shape memory characteristic [12].

#### 3.2.2. Superelasticity: Phase Change Due to Mechanical Stress

The transition from the austenitic to martensitic phase can also occur as a result of the application of mechanical stress such as that experienced by endodontic instruments during canal preparation. In most metals, when an external force causes a certain amount of slippage in the network of atoms, a permanent deformation occurs. However, in NiTi alloys, a martensitic transformation is allowed instead of this slip (Figure 4). This phase change, from austenitic to martensitic during deformation, is responsible for the superelastic behaviour. The superelasticity of NiTi allows it to deform up to 8% with the deformation remaining recoverable.

### 3.3. Advances in the NiTi Alloy

Great improvements in the mechanical properties of NiTi instruments have been achieved through subtle modifications to the ratio of the two elements. However, the key step was to develop thermomechanical treatments to adjust the transition temperatures of the alloys (Ms, Mf, As, Af). Thus, controlling the characteristics of its microstructural phases and influencing its mechanical behaviour, gradually moving away from technologies to reduce surface defects [6,13]. While the conventional NiTi alloy is in the austenitic phase during clinical use [5], the thermomechanically treated NiTi alloys contain variable amounts in the R phase and in the martensitic phase under clinical conditions [14]. These modifications lead to more flexible endodontic files with a high resistance to fracture. There are several alloys that have emerged over the years thanks to the heat treatment of NiTi:

M-WIRE (MEMORY-WIRE) (2007): Not completely composed of austenite phase under clinical conditions, but rather has small amounts of martensite and R phase [15]. M-Wire exhibits greater flexibility, greater resistance to cyclic fatigue and better mechanical properties compared to conventional NiTi [15,16].

R PHASE (2008) SybronEndo (Orange, CA, USA): Used to manufacture the Twisted File (TF) system. Because the modulus of elasticity of the R phase is less than that of austenite and martensite, an instrument made with this alloy will be more flexible than a conventional NiTi [17]. Twisted Files are more resistant to cyclic fatigue than conventional NiTi files [6].

CM-WIRE (CONTROL MEMORY-WIRE) (2010) DS Dental (Johnson City, TN, USA): All this indicates that CM-Wire is easily deformable under mechanical stress but will regain its original shape after heating in an autoclave, up to the point where an inverted deformation occurs where it will continue to deform after sterilization, which is an indication that it should be discarded. Therefore, CM-Wire instruments have been found to have greater flexibility and resistance to cyclic fatigue than conventional M-Wire and NiTi instruments [18].

BLUE AND GOLD: The distinctive colour of these alloys is due to the fact that after the heat treatment of the alloy, a layer of titanium oxide remains covering the surface of the instrument. The thickness of this layer in the Blue alloy is 60–80nm, while in the Gold alloy it is 100–140nm [19].

These instruments also exhibit controlled memory and can be deformed [20]. The main difference between CM Wire and the Blue and Gold heat treatment instruments is that these instruments are machined before going through heat treatment [21]. All Gold and Blue heat-treated files demonstrated greater flexibility and fatigue resistance compared to conventional NiTi and M-Wire alloys [22,23]. However, when compared with alloys that control the memory of the material, such as CM-Wire, the resistance to cyclical fatigue of the latter is higher [24].

EDM HYFLEX (2016) Coltene (Coltene/Whaledent, Altstätten, Switzerland): Hyflex EDM files (Coltene/Whaledent, Altstätten, Switzerland) are manufactured with CM-Wire alloy but using electric discharge machining (EDM) technology [25]. EDM is a thermal erosion process used with electrically conductive materials that result in a crateriform surface finish on the instrument [26]. HyFlex EDM exhibits significantly higher fatigue resistance compared to Hyflex CM, M-Wire and conventional NiTi [27,28], while its flexibility remains similar to that of CM Wire instruments [27,28]. In a study comparing Hyflex EDM files with ProTaper Universal and ProTaper Gold files, Kaval et al. [29] conclude that the former are more resistant to cyclical fatigue, but that ProTaper Gold are more resistant to torsion, hence they recommend using each instrumentation system on a specific type of root canals.

MAXWIRE (Martensite–Austenite–electropolish–fileX) (2016) (FKG): the first NiTi alloy that combines the effect of shape memory and superelasticity in endodontics. These instruments are in a martensitic state at room temperature and change to a curved shape when exposed to intraduct temperature due to the fact that they transform to an austenitic phase. Their curved shape allows complex canals to be prepared, with the potential to accommodate irregularities in the canal. Their mechanical action together with the agitation of the irrigant, promotes greater bacterial reduction and elimination of biofilms [30].

### 3.4. Advances in Designs

Within the design of the instrument, there are several parameters that condition its mechanical behaviour:−MASS: In their study, Turpin JL et al. [31] calculated the surface area of the cross-section of Hero (triple helix) and Profile (triple U) files and compared the stresses generated in bending forces. With the diameters of both files the same, the area for Hero instruments was approximately 30% greater than that of Profile. This larger section determined a lower flexibility in Hero compared to Profile, which would partly explain its lower resistance to cyclical fatigue [32]. Therefore, the minimal amount of core that Profile instruments have is what improves their flexibility and resistance to cyclical fatigue.−GEOMETRY: The presence or absence of radial lands or cutting edges will also influence the flexibility of the files. Beruttí et al. [7], observed comparing Profile and Protaper, that with the same diameter, the Protaper cross-sectional area was around 30% greater than that of Profile. This means that the Profile instrument has less mass than Protaper and therefore will be more elastic. However, although it was more flexible, the stress distribution was far more irregular in Profile than in Protaper, accumulating high voltage peaks in the grooves between the radial lands. Therefore, the Protaper instrument, despite having more mass and being less elastic, is capable of distributing stresses more evenly throughout the entire file, thanks, among other things, to its cross-sectional geometry.

Over time, the radial lands have been alleviated in some cases and in others they have become active cutting edges, requiring fewer instruments to complete preparation. However, we must bear in mind that the sharp edges penetrate the dentin and produce high stresses on both the instrument and the canal. In addition, they can produce an apical driving force known as the screwing effect. This is a tactile sensation that the file is rotating inside the canal in an apical direction, which can cause involuntary overextension of the instrument beyond the apical foramen [33], weakening the root and conditioning the success of the treatment. In addition, it can lead to the instrument locking inside the canal, increasing the torsional stress of the file, with its consequent risk of fracture [34].

−PITCH: a third aspect that can influence flexibility is the number of turns. The higher the pitch or fewer turns the instrument has, the more flexible it will be, but in turn, the helical angle will decrease, and the file will have less cutting capacity, so it will need more rotations to be effective. For example, the Mtwo instrument has a S-italic cross-section, with two cutting edges and a large groove between them. This type of geometry aims to reduce the mass of the core and thereby increase its flexibility. However, it has a lower number of turns per unit length, which may have led Tripi et al. [32] to find it is less resistant than K3 or Profile, despite the greater section area of the latter.−TAPER: The arrival of the ProTaper system (DENTSPLY Tulsa Dental Specialties) on the market, was a revolution. This system did not stand out for its cross-section but for its variable taper along the file, which makes each file work in a specific part of the canal and offers a shorter sequence of files. This type of design serves to minimise the contact of the file with the canal, reducing the possibility of a conical locking or screw effect, while increasing its effectiveness. Furthermore, it strategically improves flexibility and is more conservative with coronal dentin than if it had a fixed taper.−AXIS OF ROTATION: In most conventional NiTi files, the axis of rotation corresponds to the geometric centre of the cross-section. However, one of the latest advances in file design was the introduction of an off-centre transverse design. In these instruments, the geometric centre of the cross-section does not coincide with the centre of rotation. With this idea, the manufacturers affirm that the stresses during rotation are reduced together with the screwing forces, because the contacts of the instrument with the tooth are reduced and the space for detritus removal is increased [35,36]. These off-centre section files create an asymmetrical rotary motion because the instrument does not touch the canal walls constantly. Furthermore, a file with these characteristics is capable of cutting more dentin than a similarly sized file, but with a symmetrical cross-section centred on the axis of rotation. The clinical advantage of this is that a smaller and therefore more flexible file can cut the same number of teeth as a larger and more rigid one.

### 3.5. Advances in Rotation: Continuous vs. Reciprocating

It was in 2008 when Yared [37] first proposed a new approach using a single reciprocating NiTi instrument. Reciprocating rotation is an evolution of the balanced forces technique of Roane et al. [38]. Two years later, De-Deus et al. [39], based on Yared’s principle, analysed the resistance to cyclical fatigue of the Protaper Universal F2 instruments, using them in continuous rotation at 250 and 400 rpm and in reciprocating motion. They concluded that reciprocating motion lengthens the life of the instruments and that the speed at which they are used is a determining factor in the fracture time.

From here on, many authors have studied this type of rotation. Initially, two reciprocal rotation systems were introduced to the market: Reciproc (VDW, Munich, Germany) and WaveOne (Dentsply Maillefer). The main difference in these instruments is that they have a counterclockwise cutting direction, so they actively cut if the counterclockwise rotation is greater than the clockwise [40]. This is due to its cross-sectional design, in which the cutting angle with the canal walls is negative and becomes positive if the file is turned counterclockwise. However, with the exception of these instruments, all instruments are designed to cut clockwise, meaning they have a positive cutting angle. Therefore, all instruments designed for continuous clockwise rotation can also work with reciprocating movement, as long as in this movement the clockwise rotation is broader than the counterclockwise movement.

On the market there are many motors, some that include the alternative movement option and some that do not. There are two types of motors with a reciprocating rotation option [40]:−Open, which allow the angles of clockwise and counterclockwise rotation and their speed to be modified, such as: ATR Vision (ATR, Pistoia, Italy), iEndo Dual (Acteon, Merignac, France), and SAF system pro (ReDent-Nova, Ra’nana, Israel).−Closed, which are programmed and do not allow any modification. Therefore, clockwise cutting files with reciprocating movement cannot be used in these motors, since the instrument will not cut or penetrate the canal. In these motors, if we want to use the reciprocating option, we will have to use files specifically designed to cut counterclockwise, not just any type of file will work. Some examples would be: WaveOne (Dentsply, Maillefer), VDW Silver Reciroc (VDW, Munich, Germany), Motor elements (SybronEndo), and ATR Technika (ATR, Pistoia, Italy).

Evidently the industry itself sometimes limits us, therefore it is essential to know all these characteristics so that we are able to choose how we want to work.

Main differences between continuous and reciprocating rotation:

RESISTANCE TO CYCLICAL FATIGUE: De-Deus et al. [39], Gambarini et al. [41], Pedull et al. [42], Pérez-Higueras et al. [43], Kiefner et al. [44], and Vadhana et al. [45] are some of the authors who used the same file system applying different kinematics to solely evaluate the effect of the movement. In this way they can be compared without other influence variables. All of them concluded that the reciprocating movement has a statistically significant greater resistance to cyclical fatigue.

CANAL TRANSPORTATION: In this case there is more disagreement, but the systematic review by Ahn et al. [45] concludes that continuous rotation instruments may exhibit a better centring capacity due to the use of a gradual sequence of files, while reciprocating instruments only use one instrument to prepare the entire canal [46].

APICAL EXTRUSION OF DETRITUS: the literature is highly conflicting on which type of rotation expels more detritus. There are studies that conclude that continuous rotation extrudes more [47] due to the greater number of instruments used. However, some conclude that reciprocal rotation extrudes more [48] because continuous rotation improves coronal expulsion of detritus [49], while others show no differences between the two [49]. Therefore, more research is needed in this regard [45].

DENTINARY DEFECTS OR CRACKS: There is also disagreement on this point. There are studies that find fewer defects using alternating motion [50]. Some studies find the opposite [51] and others find no significant differences between movements [52]. However, recently, De-Deus et al. [53] carried out a study using fragments of cadaver jaw bones that contained three to five teeth each. A total of 178 teeth were analysed in which they made a total of four micro-CTs: pre-extraction, post-extraction, post-storage of the teeth after 3 months of dehydration, and finally after refitting them in the maxillary bone. They observed that in the first of the micro-CTs, when the tooth is in the socket, there are no dentin microcracks. Most microcracks appear after storage of the teeth. Therefore, this casts doubt on most microcrack studies since these dentin defects may have been present prior to endodontic instrumentation for other reasons unrelated to instrumentation [53].

### 3.6. Influencing Factors in the Fracture of Endodontic Files

#### 3.6.1. Anatomy of the Canal Factors

Flexion fracture in rotary NiTi instruments has been shown to occur at the point of maximum flexion, which corresponds to the point of greatest curvature of the canal. In 1997, Pruett [54] described the radius of curvature and angle of curvature concepts. Multiple studies have shown that when the angle of curvature increases and/or the radius of curvature decreases, the number of rotation cycles required for the file to fracture is reduced [54,55]. This is supported by the fact that most instruments fracture in the apical third of the canal, which is usually the area of maximum curvature and the smallest diameter of the canal [56].

#### 3.6.2. Instrument Factors

The factors of the instrument would be those already mentioned previously: manufacturing process and alloys, instrument design, and rotation.

#### 3.6.3. Forms of Use

Dynamics, technique, and experience of the operator can determine the resistance of the instruments.

##### Dynamics

(a)TORQUE: Controlled torque of electric motors is generally recommended for the use of NiTi systems. Controlled torque produces a lower yield strength than the file and reduces instrument fracture caused by torsional overload [57]. However, a clinical study investigated three levels of torque control (high, moderate, and low) during canal preparation. It noted that if the operator was inexperienced, the fracture rate decreased with low torque [58]. However, when the operator was an expert, no differences were observed using high or moderate torque.(b)ROTATION SPEED: There is disagreement in this aspect in the literature. Some studies show that speed does not influence the incidence of fracture [54,59], while others show the opposite [60]. However, manufacturers generally recommend a specific number of rotations per minute (RPM) to use instruments safely, which is typically 250–600.

##### Technique

There are multiple instrumentation techniques that have been developed over time. The crown-down technique (widening the coronal part of the canal before apical preparation), for example, has been shown to reduce torsional stress, particularly in smaller instruments [61].

Another fundamental technique when preparing a canal, regardless of the rotary instrumentation system used, is the creation of a glide path. The glide path is described as a smooth and centred preparation of the root canal from its entrance to the apical foramen [62]. It helps rotary nickel–titanium (NiTi) instruments work safely during instrumentation [63] and prevents complications such as locks, steps, transport, fractured instruments [62,64], and extrusion of detritus [65,66]. Glide paths have been shown to be able to drastically reduce torsional stress because the canal widens to at least the diameter of the tip of the first rotary instrument [64].

##### Operator Skills and Experience and Canal Preparation Time

Operator experience is a key factor in file fracture when other factors (instrument or canal morphology) remain constant [67]. It influences not only the quality of treatment, but also the time spent [68,69]. If we analyse the quality of the treatment, we agree with several authors [67,70] that inexperienced operators create more deformations and fractures in NiTi instruments than expert operators. Mandel [67] associates this fact with the time that the files are working inside the canal. Each rotation in the bend of a canal subjects the instrument to a cycle of tension–compression stress that is the most destructive factor in the load cycle. Therefore, since inexperienced operators often require more time to perform instrumentation, they are likely to cycle the instrument through more rotation cycles, thereby increasing the probability of fracture. However, there are also other factors such as the mode of use or the pressure exerted.

#### 3.6.4. Other Factors

STERILIZATION: There is conflict in the literature over the impact of sterilization on NiTi instruments, with heterogeneous methodologies and results. From the onset and propagation of cracks in the instruments [71], some find that they do not affect the incidence of fracture [72,73] while others find an increase in resistance [74], probably related to the use of new heat-treated alloys.

IRRIGANTS: It has been reported that the corrosion effect of sodium hypochlorite (NaOCl) as an irrigant can have a negative impact on the mechanical properties of NiTi instruments [75].

NUMBER OF USES: Manufacturers recommend that files should be single use. The literature is not clear when it comes to providing guidance on the number of uses. Several studies claim that the failure of NiTi instruments is more influenced by the way they are used than by the number of times they are used [76]. However, regardless of the way in which they are used, NiTi files show less resistance to flexural fatigue with repeated use, and the torque required for them to fracture is significantly less on used instruments than on new instruments [59].

REPLIKA-LIKE SYSTEMS: Moreover, in the last years, several companies worldwide started to produce NiTi instruments with similar characteristics to well-known brand systems without clear reports on production control quality or international certification. These instruments, although they are different brands, present similar characteristics to the original ones, such as the number/sequence of instruments, nomenclature, and identification (colour coding); they are named replicalike systems. It is important to know that clinical safety and efficiency of most of these systems were not confirmed yet from a scientific perspective, or the data are scarce compared with the original brand counterparts [77].

## 4. Conclusions

Therefore, when carrying out endodontic treatment we must take into account all these aspects: both all the characteristics of the instrument we are using and the characteristics of the tooth we are treating, as well as knowing our own limitations. In this way, we will reduce the number of errors as far as possible, and we will be successful in our treatments.

## Figures and Tables

**Figure 1 medicina-58-00719-f001:**
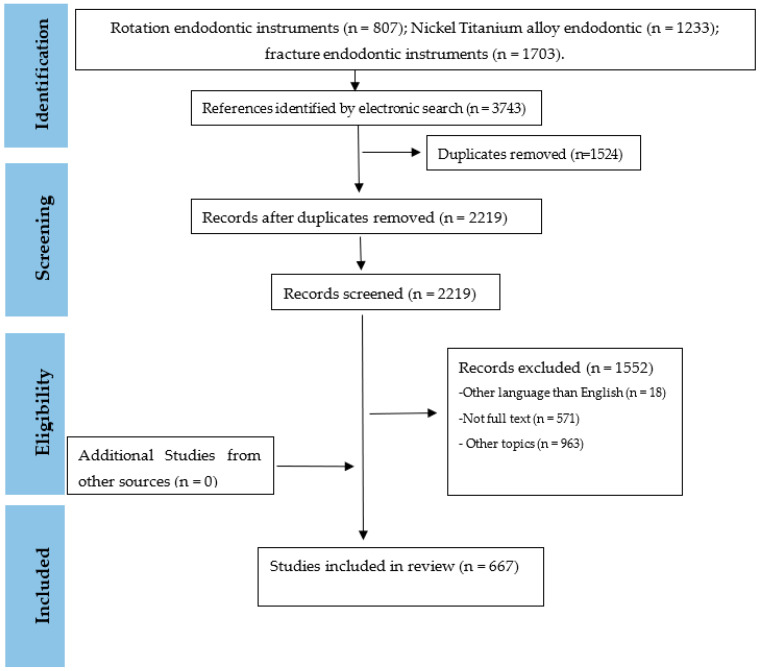
PRISMA flow diagram for review of the literature.

**Figure 2 medicina-58-00719-f002:**
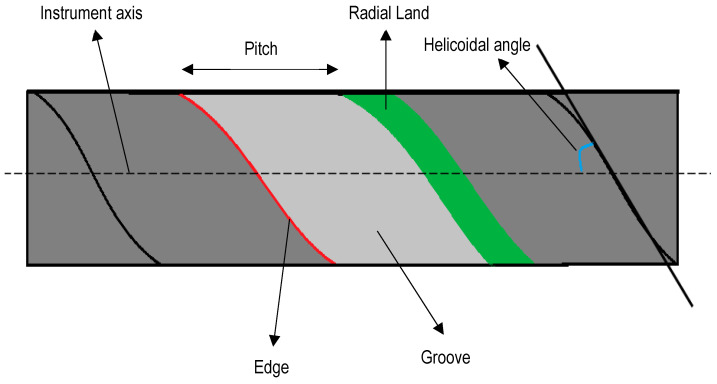
Parts of rotary file.

**Figure 3 medicina-58-00719-f003:**
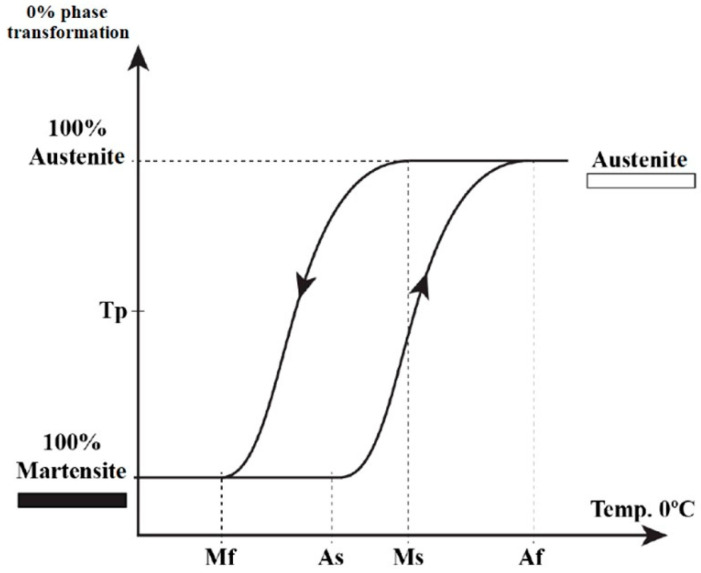
Thermogram of martensitic transformation temperatures. At a high temperature (100 °C) the NiTi alloy is in the austenitic phase and if it is cooled, alloy change to the martensitic phase (Ms) up to a certain temperature at which all the crystals of the alloy have become martensitic (Mf). The same happens if the temperature increases, such that the temperature at which this phenomenon begins is called the austenite transformation start temperature (As) and the temperature at which this phenomenon is complete is called the austenite finish temperature (Af).

**Figure 4 medicina-58-00719-f004:**
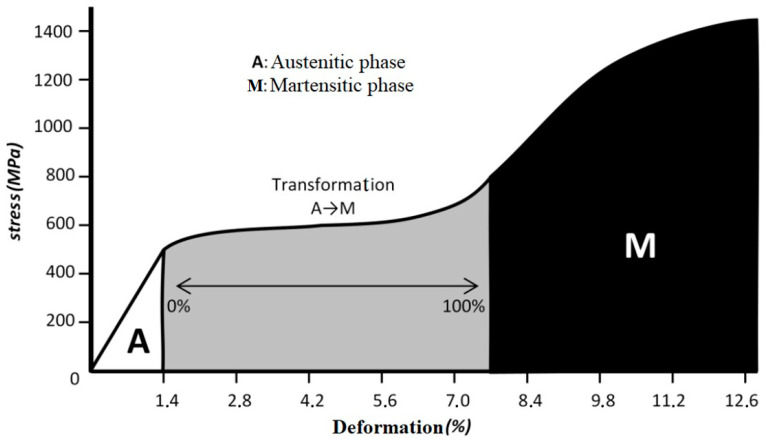
NiTi stress–strain graph. A: deformation of the austenitic phase (proportional elastic deformation). A-M: transition zone from austenitic (A) to martensitic (M), a large amount of deformation occurs without a large increase in stress. M: follows the typical behaviour of a stress–strain graph: linear elasticity, yield point and fracture point.

## Data Availability

Not applicable.

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
