# Peer review of "Endodontic Rotary Files, What Should an Endodontist Know?"

_medicina, 2022, doi:10.3390/medicina58060719_

Round 1

Reviewer 1 Report

In this manuscript, the Authors conducted a narrative review to summarize the characteristics of endodontic rotary files that should be taken into account when choosing one instrument over another in daily practice.

As a search conducted on any scientific search engine can easily reveal, there are several reviews in the literature, both narrative and systematic, that address this topic by developing it about, for example, the torque of the aforementioned tools (10.1111/eos.12802), their fracture (10.1111/aej.12484), or the characteristics of the latest devices available on the market (10.14744/eej.2019.80664). 
And this, just to mention the articles produced in the last 3 years.
Furthermore, in the present form, the manuscript presents major flaws. 
Please find below my considerations.

Abstract
1. “ignorance about the instruments…”. This sentence needs to be rephrased.
2. In the abstract is rightly presented a section Methods that instead is absent in the paper. It should be noted that although unlike systematic reviews that benefit from guidelines such as PRISMA (Preferred Reporting Items for Systematic Reviews and Meta-Analyses) statement, there are no acknowledged guidelines for narrative reviews, a presentation of the methodology used to select the reported articles is always a good rule to follow.
This allows the reader to understand the period of reference, the difference with previous reviews on the same subject, the seriousness of the research criteria, the number of researchers involved, etc.
In brief, the literature search (the ‘Methods’) is a critical step in determining the selection bias. 

Introduction
3. The Introduction appears weak. What limitations are present in previously published reviews that prompt authors to try to fill any gaps in the literature?
4. The Authors state that "there is a consensus among researchers stating that the main disadvantage of rotary NiTi systems is the unexpected fracture of the instruments” and this could be a consequence of “ignorance of the characteristics of the instruments”, hence the purpose of the review to present the characteristics of rotary instruments on the market. But, as said before, this is a topic already strongly developed.

Methods
5. Between the Introduction and Main text sections, the above-mentioned Methods are missing. 
The Methods section is not mandatory for Narrative Reviews, but if included, it adds clarity to the key messages of the NR. So either remove the Materials and Methods section from the abstract, or, as would be more appropriate, add the Methods section to the manuscript.
6. What is the search strategy (databases, keywords, etc.)
7. Which are the inclusion/exclusion criteria (types of studies, languages, time periods, others)?

Main text
8. Once the research methodology has been clarified and presented, the text should be structured according to it.

Finally, the Authors need support to improve the quality of their English throughout the manuscript.

Author Response

Abstract
1. “ignorance about the instruments…”. This sentence needs to be rephrased.

Thank you, it was changed
2. In the abstract is rightly presented a section Methods that instead is absent in the paper. It should be noted that although unlike systematic reviews that benefit from guidelines such as PRISMA (Preferred Reporting Items for Systematic Reviews and Meta-Analyses) statement, there are no acknowledged guidelines for narrative reviews, a presentation of the methodology used to select the reported articles is always a good rule to follow.
This allows the reader to understand the period of reference, the difference with previous reviews on the same subject, the seriousness of the research criteria, the number of researchers involved, etc.
In brief, the literature search (the ‘Methods’) is a critical step in determining the selection bias. 

Ok, we change the format of the abstract and added the method on the paper.

Introduction
3. The Introduction appears weak. What limitations are present in previously published reviews that prompt authors to try to fill any gaps in the literature?

We added more reasons on lines 36 37 and 38
4. The Authors state that "there is a consensus among researchers stating that the main disadvantage of rotary NiTi systems is the unexpected fracture of the instruments” and this could be a consequence of “ignorance of the characteristics of the instruments”, hence the purpose of the review to present the characteristics of rotary instruments on the market. But, as said before, this is a topic already strongly developed.

It was changed

Methods
5. Between the Introduction and Main text sections, the above-mentioned Methods are missing. 
The Methods section is not mandatory for Narrative Reviews, but if included, it adds clarity to the key messages of the NR. So either remove the Materials and Methods section from the abstract, or, as would be more appropriate, add the Methods section to the manuscript.
6. What is the search strategy (databases, keywords, etc.)
7. Which are the inclusion/exclusion criteria (types of studies, languages, time periods, others)?

All it was added.

Main text
8. Once the research methodology has been clarified and presented, the text should be structured according to it.

Reviewer 2 Report

The manuscript consists on a narrative review of the importance of NiTi rotary files. Although the topic is by far not new, it appears interesting.

The revision makes a good overview of the most relevant points on the issue.

Title looks ok

The authors should notice that the Abstract in this journal does not necessary requires to be structured. Therefore I would recommend the elimination of the subheading since this is a review that did not follow a traditional experimental model with methods.

It is very important for the authors to use correct scientific terms. When the authors start a manuscript with the word “Ignorance” I wonder who is ignorant? The clinicians that are not aware of things or that authors that are not aware of scientific term. Please change the sentence “ignorance about the instruments they are using on the part of 11 clinicians who perform endodontics is all too frequent today” for something like “Clinicians should be aware of all the characteristics and capacities of the instruments that are possible to be used when conducting a root canal treatment”. Feel free to adjust

The Abstract should be adjusted. The authors are conducting a Narrative review. They do not need to conduct an experimental like abstract with a “Material and Methods” section saying that you have reviewed papers. Obviously you did. Please reformulate the Abstract in a way that it looks like a narrative review abstract leaving a key about the main takeaways including some characteristics that you feel are relevant.

The keywords should be in alphabetic order.

Introduction, first sentence… it is not “steel” but stainless-steel

Which ISO regulations? Citation

Regarding the sentence: “Currently, there is a consensus among researchers stating that the main disadvantage of 31 rotary NiTi systems is the unexpected fracture of the instruments”… try to look it from the good side also… “please change to “advantages and disadvantages”

Regarding the sentence: “However, many of the fractures that occur in daily clinical practice may be the result of ignorance of the characteristics of the instrument with 34 which one is working.”… try not the call others dummies ok? Remove all the “ignorance” and derivates from the text and pass the some information on a polite manner.

Rationale and aim sentence looks ok

Remove the “Main Text” subheading. Not needed. But if you want if badly… use the usual “Review” sub-heading

Regarding the “(Cohen & Hargreaves 2011)”… is this a reference? If yes… it should be numbered.

Do the authors have a copyright authorization to re-publish the Figure 1? If not I suggest making your one which is not difficult.

In the “Shape memory: phase change by temperature”, what is the meaning of TTR and RTTR?

Regarding the “ADVANCES IN ROTATION: CONTINUOUS VS RECIPROCATING” subheading. The authors are not considering a relevant point. The oscillatory asymmetric motion may be counter-clockwise (reciprocation motion) or clockwise, which in the latter case may be used in rotary files. I suggest adding this information and debate the paper DOI: 10.1016/j.joen.2020.05.001

Regarding this issue:

“2.- INSTRUMENT FACTORS: (previously described) 316

MANUFACTURING PROCESS and ALLOYS 317

INSTRUMENT DESIGN 318

ROTATION”

I understand you do not wish to debate it again, but a short sentence with the main key takeaways would be welcome. Placing this this way is not acceptable in a scientific work.

In the “4.- OTHER FACTORS” I recommend the authors to add a new trend coming recently on endodontics. Which are the “replica-like instruments”… the clinicians should be aware of their existence and they although they are 100% legal, they do nor represent a perfect copy of the original ones. Additionally, the clinicians must be aware of the existence of counterfeict instruments, which are illegal and unsafe. Please check out and debate these two update and highly relevant studies DOI: 10.1016/j.joen.2020.08.021 and DOI: 10.1111/iej.13463

If the authors wish to add some updated, this issues are mandatory and novel to endodontics

The reference list looks acceptable and updated.

Author Response

The authors should notice that the Abstract in this journal does not necessary requires to be structured. Therefore I would recommend the elimination of the subheading since this is a review that did not follow a traditional experimental model with methods.

Perfect, thank you, we already removed.

It is very important for the authors to use correct scientific terms. When the authors start a manuscript with the word “Ignorance” I wonder who is ignorant? The clinicians that are not aware of things or that authors that are not aware of scientific term. Please change the sentence “ignorance about the instruments they are using on the part of 11 clinicians who perform endodontics is all too frequent today” for something like “Clinicians should be aware of all the characteristics and capacities of the instruments that are possible to be used when conducting a root canal treatment”. Feel free to adjust

Thank you for interesting aportation, it was changed.

The Abstract should be adjusted. The authors are conducting a Narrative review. They do not need to conduct an experimental like abstract with a “Material and Methods” section saying that you have reviewed papers. Obviously you did. Please reformulate the Abstract in a way that it looks like a narrative review abstract leaving a key about the main takeaways including some characteristics that you feel are relevant.

We added this.

The keywords should be in alphabetic order 

We think they are already in alphabetical order

Introduction, first sentence… it is not “steel” but stainless-steel

Changed

Which ISO regulations? Citation 

Added

Regarding the sentence: “Currently, there is a consensus among researchers stating that the main disadvantage of 31 rotary NiTi systems is the unexpected fracture of the instruments”… try to look it from the good side also… “please change to “advantages and disadvantages” 

Ok, we changed

Regarding the sentence: “However, many of the fractures that occur in daily clinical practice may be the result of ignorance of the characteristics of the instrument with 34 which one is working.”… try not the call others dummies ok? Remove all the “ignorance” and derivates from the text and pass the some information on a polite manner.

Sorry, we didn't mean to be rude

Rationale and aim sentence looks ok 

Remove the “Main Text” subheading. Not needed. But if you want if badly… use the usual “Review” sub-heading

Changed

Regarding the “(Cohen & Hargreaves 2011)”… is this a reference? If yes… it should be numbered.

Changed

Do the authors have a copyright authorization to re-publish the Figure 1? If not I suggest making your one which is not difficult.

Changed

In the “Shape memory: phase change by temperature”, what is the meaning of TTR and RTTR?

Changed

Regarding the “ADVANCES IN ROTATION: CONTINUOUS VS RECIPROCATING” subheading. The authors are not considering a relevant point. The oscillatory asymmetric motion may be counter-clockwise (reciprocation motion) or clockwise, which in the latter case may be used in rotary files. I suggest adding this information and debate the paper DOI: 10.1016/j.joen.2020.05.001

We think that it is considering o lines: 294-314

Regarding this issue:

“2.- INSTRUMENT FACTORS: (previously described) 316

MANUFACTURING PROCESS and ALLOYS 317

INSTRUMENT DESIGN 318

ROTATION”

I understand you do not wish to debate it again, but a short sentence with the main key takeaways would be welcome. Placing this this way is not acceptable in a scientific work. 

Thank you for the aportation, it was changed.

In the “4.- OTHER FACTORS” I recommend the authors to add a new trend coming recently on endodontics. Which are the “replica-like instruments”… the clinicians should be aware of their existence and they although they are 100% legal, they do nor represent a perfect copy of the original ones. Additionally, the clinicians must be aware of the existence of counterfeict instruments, which are illegal and unsafe. Please check out and debate these two update and highly relevant studies DOI: 10.1016/j.joen.2020.08.021 and DOI: 10.1111/iej.13463

If the authors wish to add some updated, this issues are mandatory and novel to endodontics 

Thank you, it was added

The reference list looks acceptable and updated.

Round 2

Reviewer 1 Report

The authors need support to improve the quality of their English throughout the manuscript, but since these are errors that are easily eliminated during the copy-editing stage, it will be the journal editors' concern to take care of this.

Reviewer 2 Report

Dear authors, I have no more concerns. Thank you.